**Data Availability Statement:** All DPC datasets have ethical or legal restrictions for public

# Comparison of short-term clinical outcomes between low-dose prasugrel and clopidogrel as part of triple antithrombotic therapy in patients requiring oral anticoagulant therapy and percutaneous coronary intervention

Hideki Kitahara[1]*, Kazuya Tateishi[1], Yuki Shiko[2], Yusuke Inaba[2], Yoshio Kobayashi[1], Takahiro Inoue[3]

1 Department of Cardiovascular Medicine, Chiba University Graduate School of Medicine, Chiba, Japan, 2 Biostatistics Section, Clinical Research Center, Chiba University Hospital, Chiba, Japan, 3 Healthcare Management Research Center, Chiba University Hospital, Chiba, Japan

* hidekitahara0306@gmail.com

## Abstract

### Background

Triple antithrombotic therapy, including dual antiplatelet therapy and oral anticoagulant (OAC), is recommended for a short-term period after percutaneous coronary intervention (PCI) in patients requiring anticoagulation therapy. The purpose of this study was to compare in-hospital clinical outcomes between low-dose prasugrel (3.75 mg/day) and clopidogrel, as part of triple antithrombotic therapy, using a large database in Japan.

### Methods

Patients with ischemic heart disease who underwent PCI between January 2015 and December 2019, and were prescribed triple therapy with aspirin, a P2Y12 inhibitor (clopidogrel or low-dose prasugrel), and OAC (direct oral anticoagulant: DOAC or vitamin K antagonist: VKA), were selected from the Diagnosis Procedure Combination database. The primary outcome was in-hospital mortality. The secondary outcomes were myocardial infarction, ischemic stroke, bleeding stroke, gastrointestinal bleeding, and blood transfusion.

### Results

Overall, 5,777 patients were eligible in this analysis. The patients were divided into 4 subgroups according to the type of P2Y12 inhibitor and OAC: clopidogrel/DOAC (n = 1,628), clopidogrel/VKA (n = 1,334), prasugrel/DOAC (n = 1,607), and prasugrel/VKA (n = 1,208). There was no significant difference in the incidence of death and gastrointestinal bleeding among the 4 subgroups. The prasugrel/DOAC group had significantly lower incidence of MI (OR 0.566, 95% CI 0.348–0.921). The incidence of ischemic stroke was significantly lower

deposition due to inclusion of sensitive information from the human participants. Thus, all enquiries should be addressed to the data management committee via e-mail: byoin-kikaku@chiba-u.jp.

**Funding:** The authors received no specific fundings for this work.

**Competing interests:** The authors have declared that no competing interests exist.

in the prasugrel/DOAC group (OR 0.701, 95% CI 0.502–0.979), and significantly higher in the clopidogrel/VKA group (OR 1.680, 95% CI 1.273–2.216). Need for blood transfusion was less frequent in the prasugrel/DOAC group (OR 0.729, 95% CI 0.598–0.890), and more frequent in both the clopidogrel/VKA group (OR 1.424, 95% CI 1.187–1.708) and the prasugrel/VKA group (OR 1.633, 95% CI 1.367–1.950).

## Conclusions

Combination of low-dose prasugrel and DOAC was associated with lower incidence of MI, ischemic stroke, and blood transfusion. Low-dose prasugrel may be feasible as part of triple therapy in patients undergoing PCI.

## Introduction

Although triple antithrombotic therapy, which includes oral anticoagulant (OAC) and dual antiplatelet therapy (DAPT), in patients requiring anticoagulation therapy and percutaneous coronary intervention (PCI) for ischemic heart disease has become obsolete because of its high bleeding risk [1], it is still used for a short-term period after PCI with a certain frequency. In recent guidelines and expert consensus documents for antithrombotic therapy after PCI, the recommended duration of triple therapy is up to 1 week to 1 month, followed by dual antithrombotic therapy with OAC and a P2Y12 inhibitor [2–7]. It is well established that direct oral anticoagulant (DOAC) is more preferable to prevent bleeding events in combination with antiplatelet therapy compared with vitamin K antagonist (VKA) [8, 9]. However, VKA is essential to patients with mechanical heart valves or severe renal impairment with some frequency. In terms of P2Y12 inhibitor, use of standard-dose prasugrel (loading/maintenance dose: 60/10 mg/day) or ticagrelor should be avoided as part of triple therapy in the Western guidelines, because previous studies have reported greater risk of major bleeding compared with clopidogrel [10–12].

In Japan and some other Asian countries [13], on the other hand, the approved dose of prasugrel is about one-third (loading/maintenance dose: 20/3.75 mg/day) of that in Western countries. Therefore, this low-dose prasugrel may be available as part of triple therapy, since its efficacy and safety in patients undergoing PCI for acute coronary syndrome (ACS) and stable coronary artery disease (CAD) have been documented in previous studies [14, 15]. A registry-based study previously reported that triple antithrombotic therapy with low-dose prasugrel did not increase the risk of bleeding compared with clopidogrel [16]. However, there is still not enough information regarding the impact of low-dose prasugrel on clinical outcomes in patients requiring triple antithrombotic therapy based on results from large-scale analysis. Thus, the purpose of this study was to compare in-hospital clinical outcomes between low-dose prasugrel and clopidogrel, as part of triple antithrombotic therapy, using a large database in Japan.

## Materials and methods

### Data source

This study used the database of Diagnosis Procedure Combination (DPC) system, which contains administrative information acquired during hospitalization and is used to calculate reimbursements to hospitals in Japan. Data were regularly collected from voluntarily participating hospitals that operate under the DPC system. The database includes summarized inpatient

information, such as age, sex, height, weight, the most resource-consuming disease, other major diagnoses and comorbidities, secondary disease, prescribed medications, treatment procedures, transfusion, in-hospital death, and other hospital-related information. This study was approved by the ethics committee of Chiba University Hospital (unique identifier: 3309). The requirement for informed consent was waived because of the anonymous nature of the data.

## Study population

The study flowchart of patient enrollment and final study sample is shown in Fig 1. In this retrospective study, a total of 96,396 patients aged ≥20 years who underwent PCI at 165 hospitals between January 2015 and December 2019 were screened. The patients in whom the most resource-consuming disease during hospitalization was not identified as ischemic heart disease (ICD-10 codes: I20, I21, I22, I23, I24, and I25) based on the International Classification of Diseases (ICD-10) coding, and those with no record that antithrombotic agents were prescribed, were excluded. From 79,023 patients receiving antithrombotic therapy, any antithrombotic therapy other than triple therapy with DAPT and OAC were excluded from the present analysis. Furthermore, as a part of triple therapy, DAPT with any antiplatelet agent other than clopidogrel or prasugrel were also excluded. After the patients with duplicate prescription of the same class antithrombotic agents (e.g. both DOAC and VKA were prescribed in the same hospital stay) were excluded, 5,777 patients were finally eligible for analysis in the present study. The patients were divided into 4 subgroups according to the type of P2Y12 inhibitor (clopidogrel or prasugrel) and OAC (DOAC or VKA).

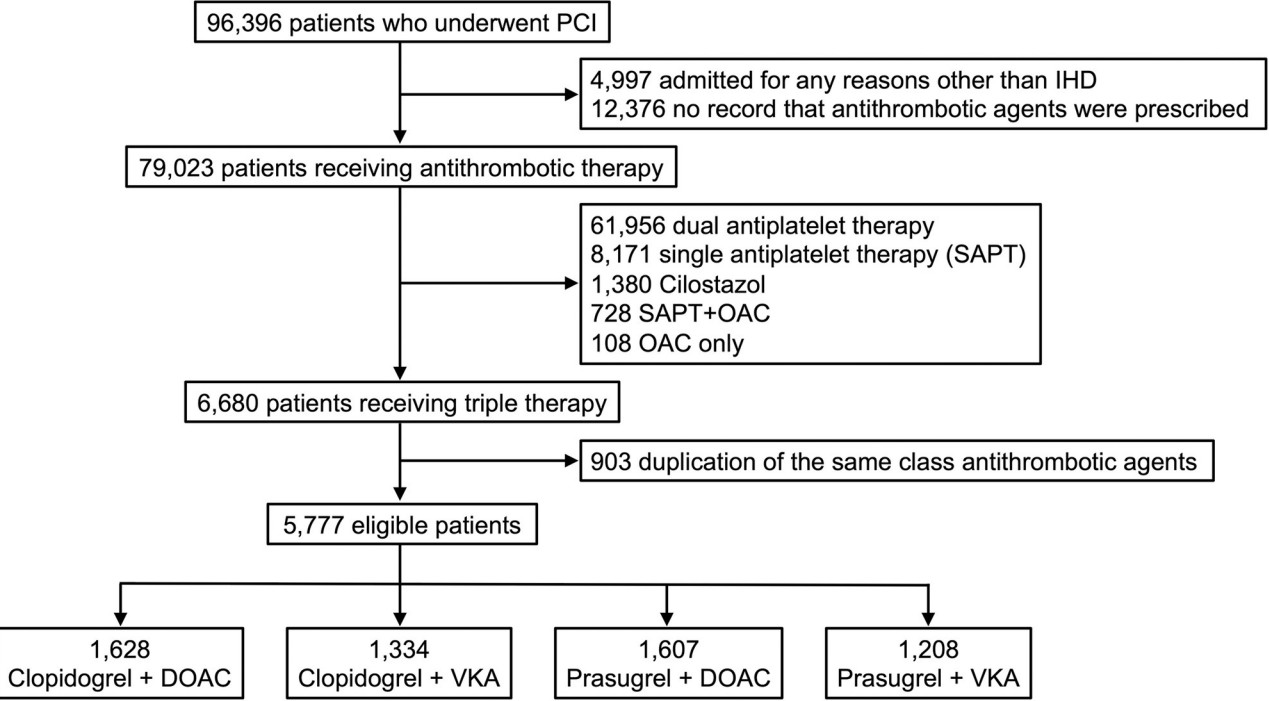

**Fig 1. Flowchart for patient selection in the present study.** DOAC: direct oral anticoagulant, IHD: ischemic heart disease, OAC: oral anticoagulant, PCI: percutaneous coronary intervention, SAPT: single antiplatelet therapy, VKA: vitamin K antagonist.

## Clinical outcomes

The primary outcome in the present study was in-hospital mortality. The secondary outcomes were in-hospital cardiovascular and bleeding adverse events, such as myocardial infarction, ischemic stroke, bleeding stroke, gastrointestinal bleeding, and blood transfusion as a surrogate marker of bleeding events.

## Baseline variables

We included the following baseline characteristics as baseline variables: age, sex, body mass index (BMI), comorbidities (hypertension, dyslipidemia, diabetes, chronic kidney disease [CKD]), the presentation of myocardial infarction, and the total duration of hospitalization. We also evaluated medications, including antiplatelet agents, anticoagulants, aspirin, statins, angiotensin converting enzyme inhibitor and angiotensin II receptor blocker (ACE-I/ARB), β-blockers, proton pump inhibitor (PPI), steroid, and non-steroidal inti-inflammatory drugs. Potassium-competitive acid blockers were included in PPI.

## Statistical analysis

Statistical analysis was performed using JMP® 14.0 (SAS Institute, Cary, NC, USA) and SAS statistical software package, v9.4 (SAS Institute, Cary, NC, USA). Categorical variables are presented as percentages and compared among the 4 subgroups using Pearson's chi-squared test. Continuous variables are presented as mean ± SD. Comparisons of continuous variables among the 4 subgroups were conducted with Analysis of variance (ANOVA). Multivariate analysis was also performed for incidence of in-hospital adverse events among the 4 subgroups. Furthermore, we compared the incidence of in-hospital adverse events between the 2 specific regimens in comparison with the clopidogrel/DOAC group as reference, and estimated odds ratios (OR) with 95% confidence interval (CI) of the treatment effects of the 2 regimens using the Firth's logistic regression analysis. These multivariate analyses were performed with adjustment for age, sex, BMI, hypertension, dyslipidemia, diabetes, CKD, statins, ACE-I/ARB, β-blocker, PPI, steroid, and non-steroidal anti-inflammatory drug, uniformly in each analysis for all patients, AMI patients, and non-AMI patients. The specific factors included for adjustment were AMI in the analysis for all patients, anterior AMI and Killip classification in the analysis for AMI patients, and unstable angina in the analysis for non-AMI patients. In terms of myocardial infarction in non-AMI patients, we adjusted for age, male, BMI, hypertension, dyslipidemia, diabetes, CKD, non-steroidal anti-inflammatory drug, and unstable angina, because of the relatively small number of the events. Since the incidence of bleeding stroke was too low, multivariate analyses were not performed for this event in the present study. A $p < 0.05$ was considered statistically significant.

## Results

Out of 5,777 patients enrolled in this study, 2,962 (51.3%) patients were prescribed clopidogrel and 2,815 (48.7%) patients were low-dose prasugrel as a P2Y12 inhibitor. Out of 2,815 patients with prasugrel, only 4 patients (0.1%) had much lower dose of prasugrel (2.5 mg/day), which is considered for patients with low body weight. As OAC, 3,235 (56.0%) patients were prescribed DOAC, and 2,542 (44.0%) patients were VKA. Consequently, patients were divided into 4 subgroups: clopidogrel/DOAC (n = 1,628), clopidogrel/VKA (n = 1,334), prasugrel/DOAC (n = 1,607), and prasugrel/VKA (n = 1,208). Comparison of baseline characteristics among the 4 subgroups are summarized in Table 1. Almost all variables, except for BMI and steroid use, were significantly different among the 4 subgroups. Briefly, the prasugrel/VKA group had the

**Table 1. Baseline clinical characteristics.**

| | Clopidogrel /DOAC (n = 1,628) | Clopidogrel /VKA (n = 1,334) | Prasugrel /DOAC (n = 1,607) | Prasugrel /VKA (n = 1,208) | p value |
|---|---|---|---|---|---|
| Age (years) | 74.2±8.8 | 72.0±10.8 | 73.0±10.8 | 69.9±11.8 | <0.001 |
| Male (%) | 77.5 | 80.8 | 77.5 | 81.1 | 0.016 |
| Body weight (kg) | 63.2±12.7 | 63.0±12.5 | 64.0±13.6 | 63.8±13.4 | 0.118 |
| BMI (kg/m$^2$) | 24.1±3.8 | 24.0±3.9 | 24.1±3.7 | 23.9±3.7 | 0.246 |
| Hypertension (%) | 63.5 | 59.8 | 69.4 | 62.3 | <0.001 |
| Dyslipidemia (%) | 59.6 | 58.5 | 65.8 | 64.2 | <0.001 |
| Diabetes (%) | 38.8 | 39.4 | 31.6 | 36.3 | <0.001 |
| Chronic kidney disease (%) | 1.0 | 5.0 | 1.2 | 3.8 | <0.001 |
| AMI (%) | 23.0 | 30.9 | 47.9 | 58.0 | <0.001 |
| Medication | | | | | |
| Aspirin (%) | 100 | 100 | 100 | 100 | na |
| Statins (%) | 75.6 | 75.8 | 81.6 | 82.4 | <0.001 |
| ACE-I/ARB (%) | 55.0 | 51.2 | 59.4 | 61.8 | <0.001 |
| β-blocker (%) | 67.3 | 70.4 | 69.9 | 74.1 | 0.001 |
| PPI (%) | 79.7 | 79.4 | 85.8 | 86.0 | <0.001 |
| Steroid (%) | 3.0 | 4.1 | 3.0 | 3.5 | 0.346 |
| NSA (%) | 12.4 | 17.0 | 13.4 | 17.7 | <0.001 |
| Hospitalization day (days) | 9.9±11.6 | 14.4±18.5 | 13.1±13.7 | 19.0±21.4 | <0.001 |

ACE-I: angiotensin converting enzyme inhibitor, AMI: acute myocardial infarction, ARB: angiotensin II receptor blocker, BMI: body mass index, DOAC: direct oral anticoagulant, NSA: non-steroidal anti-inflammatory drugs, PPI: proton pump inhibitor, VKA: vitamin K-antagonist.

youngest patient age compared with other 3 groups. There were more males, CKD, and prescription of NSA in the VKA groups, regardless of the type of P2Y12 inhibitor. While the prasugrel groups had more dyslipidemia, AMI presentation, and prescription of statins, ACE-I/ARB, and PPI, the clopidogrel groups showed a higher prevalence of diabetes, regardless of the type of OAC. Hospital stay was the longest in the prasugrel/VKA group compared with the other 3 groups.

The incidence of in-hospital clinical events and the results of univariate and multivariate analyses were shown in Tables 2–4. The multivariate analyses demonstrated that the incidence of ischemic stroke and blood transfusion were significantly different among the 4 subgroups in all patients (p = 0.001 and p<0.001, respectively), that the incidence of death and blood

**Table 2. Total clinical events: All patients.**

| | Clopidogrel /DOAC (n = 1,628) | Clopidogrel /VKA (n = 1,334) | Prasugrel /DOAC (n = 1,607) | Prasugrel /VKA (n = 1,208) | Univariate p value | Multivariate p value |
|---|---|---|---|---|---|---|
| Death | 14 (0.86%) | 23 (1.72%) | 33 (2.05%) | 30 (2.48%) | 0.007 | 0.175 |
| Myocardial infarction | 17 (1.04%) | 15 (1.12%) | 10 (0.62%) | 19 (1.57%) | 0.110 | 0.145 |
| Ischemic stroke | 40 (2.46%) | 47 (3.52%) | 24 (1.49%) | 21 (1.74%) | 0.001 | 0.001 |
| Bleeding stroke | 4 (0.25%) | 0 (0%) | 4 (0.25%) | 4 (0.33%) | 0.272 | - |
| Gastrointestinal bleeding | 8 (0.49%) | 13 (0.97%) | 23 (1.43%) | 18 (1.49%) | 0.026 | 0.068 |
| Blood transfusion | 63 (3.87%) | 129 (9.67%) | 87 (5.41%) | 148 (12.25%) | <0.001 | <0.001 |

DOAC: direct oral anticoagulant, VKA: vitamin K-antagonist.

**Table 3. Clinical events in non-AMI patients.**

| | Clopidogrel /DOAC (n = 1,255) | Clopidogrel/ VKA (n = 928) | Prasugrel /DOAC (n = 837) | Prasugrel /VKA (n = 508) | Univariate p value | Multivariate p value |
|---|---|---|---|---|---|---|
| Death | 3 (0.24%) | 2 (0.22%) | 2 (0.24%) | 5 (0.98%) | 0.064 | 0.013 |
| Myocardial infarction | 4 (0.32%) | 5 (0.54%) | 3 (0.36%) | 4 (0.79%) | 0.557 | 0.936 |
| Ischemic stroke | 29 (2.31%) | 30 (3.23%) | 13 (1.55%) | 8 (1.57%) | 0.074 | 0.070 |
| Bleeding stroke | 1 (0.08%) | 0 (0%) | 0 (0%) | 2 (0.39%) | 0.066 | - |
| Gastrointestinal bleeding | 6 (0.48%) | 6 (0.65%) | 9 (1.08%) | 4 (0.79%) | 0.450 | 0.369 |
| Blood transfusion | 29 (2.31%) | 61 (6.57%) | 20 (2.39%) | 39 (7.68%) | <0.001 | <0.001 |

DOAC: direct oral anticoagulant, VKA: vitamin K-antagonist.

transfusion were significantly different among the 4 subgroups in non-AMI patients (p = 0.013 and p<0.001, respectively), and that the incidence of ischemic stroke and blood transfusion were significantly different among the 4 subgroups in AMI patients (p = 0.039 and p<0.001, respectively).

The results of multivariate analyses to compare the incidence of in-hospital adverse events between the 2 specific regimens with the clopidogrel/DOAC group as reference are described in Fig 2. Regarding the incidence of death, there was no significant difference, except for the significantly higher incidence in the prasugrel/VKA group (OR 3.586, 95% CI 1658–7.756) in non-AMI patients. In terms of myocardial infarction, the prasugrel/DOAC group had the significantly lowest incidence in all patients (OR 0.566, 95% CI 0.348–0.921) and in AMI patients (OR 0.520, 95% CI 0.296–0.913), but not in non-AMI patients, in comparison with the clopidogrel/DOAC group as reference. The incidence of ischemic stroke was significantly lower in the prasugrel/DOAC group only in all patients (OR 0.701, 95% CI 0.502–0.979), and significantly higher in the clopidogrel/VKA group in all patients (OR 1.680, 95% CI 1.273–2.216), in non-AMI patients (OR 1.556, 95% CI 1.097–2.207), and in AMI patients (OR 1.637, 95% CI 1.064–2.519), compared with the clopidogrel/DOAC group. The incidence of GI bleeding was not significantly different between the 2 specific regimens in comparison with the clopidogrel/DOAC group as reference. Need for blood transfusion was less frequent in the prasugrel/DOAC group in all patients (OR 0.729, 95% CI 0.598–0.890) and in non-AMI patients (OR 0.548, 95% CI 0.372–0.805). In addition, it was more frequently performed in the clopidogrel/VKA groups in all patients (OR 1.424, 95% CI 1.187–1.708) and in non-AMI patients (OR 1.652, 95% CI 1.250–2.183), and in the prasugrel/VKA group in all patients (OR 1.633, 95% CI

**Table 4. Clinical events in AMI patients.**

| | Clopidogrel /DOAC (n = 373) | Clopidogrel /VKA (n = 406) | Prasugrel /DOAC (n = 770) | Prasugrel /VKA (n = 700) | Univariate p value | Multivariate p value |
|---|---|---|---|---|---|---|
| Death | 11 (2.95%) | 21 (5.17%) | 31 (4.03%) | 25 (3.57%) | 0.411 | 0.226 |
| Myocardial infarction | 13 (3.49%) | 10 (2.46%) | 7 (0.91%) | 15 (2.14%) | 0.025 | 0.114 |
| Ischemic stroke | 11 (2.95%) | 17 (4.19%) | 11 (1.43%) | 13 (1.86%) | 0.016 | 0.039 |
| Bleeding stroke | 3 (0.80%) | 0 (0%) | 4 (0.52%) | 2 (0.29%) | 0.300 | - |
| Gastrointestinal bleeding | 2 (0.54%) | 7 (1.72%) | 14 (1.82%) | 14 (2.00%) | 0.316 | 0.113 |
| Blood transfusion | 34 (9.12%) | 68 (16.75%) | 67 (8.70%) | 109 (15.57%) | <0.001 | <0.001 |

DOAC: direct oral anticoagulant, VKA: vitamin K-antagonist.

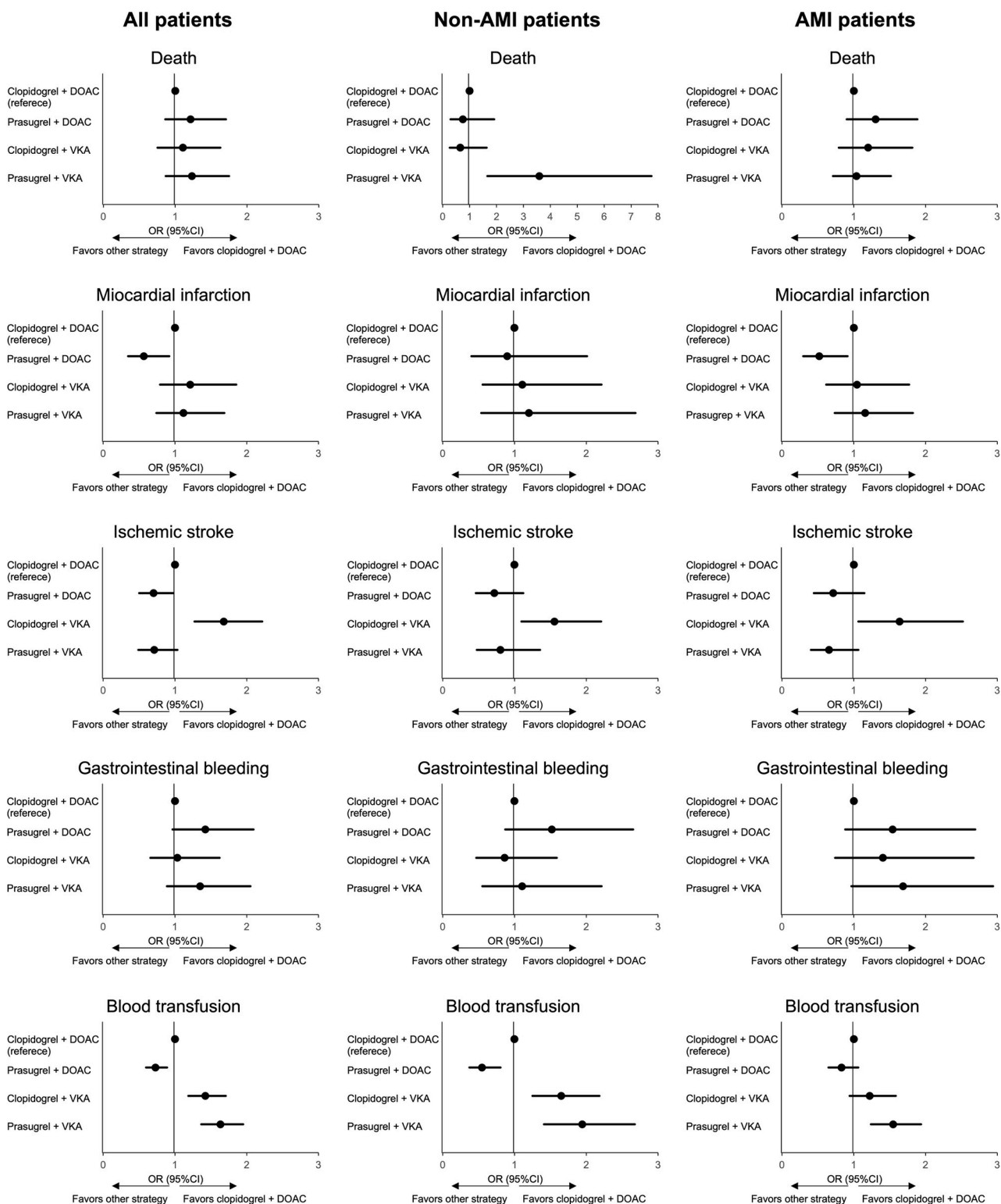

**Fig 2. Impact of the combination of P2Y12 inhibitor and OAC on in-hospital adverse events.** The incidence of in-hospital adverse events, including death, myocardial infarction, ischemic stroke, gastrointestinal bleeding, and blood transfusion, was compared between the 2 specific regimens with the clopidogrel/DOAC group as reference in all patients, AMI patients, and non-AMI patients. AMI: acute myocardial infarction, DOAC: direct oral anticoagulant, VKA: vitamin K antagonist.

1.367–1.950), in non-AMI patients (OR 1.943, 95% CI 1.411–2.676), and in AMI patients (OR 1.549, 95% CI 1.239–1.937).

## Discussion

This large-scale cohort study investigated the impact of combination of triple antithrombotic therapy on in-hospital clinical outcomes in ischemic heart disease patients undergoing PCI. Overall, the primary outcome of in-hospital death was similarly observed among the 4 subgroups stratified by the combination of a P2Y12 inhibitor (clopidogrel and prasugrel) and OAC (DOAC and VKA). The prasugrel/DOAC group demonstrated a lower rate of myocardial infarction, ischemic stroke, and blood transfusion in comparison with the clopidogrel/DOAC group as reference. The VKA groups showed a consistently higher rate of blood transfusion compared with the DOAC groups. Low-dose prasugrel and DOAC may be a balanced combination as part of triple therapy in patients requiring PCI and anticoagulation therapy.

In recent guidelines and expert consensus documents of antithrombotic therapy for patients with ischemic heart disease, dual antithrombotic therapy with a P2Y12 inhibitor plus OAC is commonly recommended in atrial fibrillation (AF) patients after PCI [2–7]. However, there are no guidelines to recommend complete dual antithrombotic therapy at the time of or immediately after PCI for every AF patient, and triple antithrombotic therapy is still basically recommended for a short-term period after PCI. In a sub-analysis of AUGUSTUS trial, a randomized controlled trial to evaluate the safety of apixaban vs. VKA and aspirin vs. placebo in AF patients after acute coronary syndrome or PCI, it was reported that adding aspirin to dual antithrombotic therapy (i.e. triple therapy) for up to 30 days led to a decrease in severe ischemic events, although the trade-off relation for an increase in severe bleeding was observed [17]. Therefore, triple antithrombotic therapy may be still needed, especially in patients with low bleeding risk and high thrombotic risk for a certain period after PCI.

It has been established that DOAC is more preferable to prevent bleeding events in combination with antiplatelet therapy compared with VKA, if patients are eligible for DOAC [8, 9]. In terms of P2Y12 inhibitor, use of standard-dose prasugrel or ticagrelor should be avoided as part of triple therapy in the western guidelines, because previous studies have reported the greater risk of major bleeding compared with clopidogrel [10–12, 18]. In Japan and some other Asian countries [13], on the other hand, the approved loading/maintenance dose of prasugrel (20/3.75 mg/day) is about one-third of that in other countries (60/10 mg/day). When considering clinical introduction of prasugrel in Japan, special emphasis was placed on determining the dose to reduce the risk of bleeding events, because Japanese patients were considered to be older and have lower weight compared with Western patients. Dose-finding studies were carefully conducted [19], and randomized controlled trials confirmed the efficacy and safety of prasugrel 20/3.75 mg/day versus clopidogrel 300/75 mg/day in ACS patients as well as stable CAD patients undergoing PCI [14, 15]. Thus, this low-dose prasugrel has been substantially used in Japan, even in patients requiring triple therapy. Although a previous study, TWMU-AF PCI Registry, reported that the incidence of ischemic and bleeding events were not significantly different between AF patients undergoing PCI who were prescribed low-dose prasugrel or clopidogrel as part of their triple therapy [16], there was not sufficient information regarding the actual distribution and the efficacy of low-dose prasugrel as part of triple therapy in clinical settings. In this large-scale cohort study, 48.7% of patients received low-dose prasugrel as part of triple antithrombotic therapy, and the feasibility of low-dose prasugrel was shown, especially in combination with DOAC.

It has been reported that standard-dose prasugrel reduces ischemic events and increases bleeding events compared with clopidogrel [20, 21]. Thus, in western countries, prasugrel is

mainly used for ACS patients at high thrombotic risk who are scheduled for PCI. Regarding low-dose prasugrel, better clinical outcomes without significant increase in bleeding events in patients undergoing PCI for both ACS and stable CAD compared with clopidogrel have been reported in previous studies [14, 15]. It is well known that the antiplatelet effect of clopidogrel is attenuated in patients with poor metabolizing phenotype of CYP2C19 [22, 23], and clopidogrel is less effective on preventing cardiovascular events after PCI in such patients compared with other phenotypes [24]. Of note, 13–23% of Asian patients are CYP2C19 poor metabolizers, which is much more frequent than other races [25, 26]. On the other hand, the effect of prasugrel is less influenced by the CYP2C19 polymorphisms, and even low-dose prasugrel reportedly achieves stronger platelet inhibition than clopidogrel in CAD patients [27–29]. Even in cases of triple antithrombotic therapy, the interindividual variability in response to clopidogrel and more consistent effect of prasugrel might affect clinical outcomes. In the present study, ischemic events (myocardial infarction and ischemic stroke) were reduced without increase in bleeding events in patients prescribed low-dose prasugrel and DOAC, although further study is warranted to address the reason why blood transfusion was reduced in this subgroup compared with other 3 subgroups. In terms of gastrointestinal bleeding, the incidence was quite low, and no significant difference was observed among the 4 subgroups. In this study population, PPI was administered in around 80–85% of patients. This high frequency of PPI administration might be associated with the reduced incidence of gastrointestinal bleeding.

In the clopidogrel/VKA group, there was a higher incidence of ischemic stroke compared with other 3 groups. In general, considering high bleeding risk, patients with VKA and antiplatelet agents more frequently have poor anticoagulation control compared with DOAC, and such poor control of VKA is associated with more adverse events, including ischemic stroke [30]. It might be one of the mechanisms of the higher incidence of ischemic stroke in the clopidogrel/VKA group, although it is unclear why the lower incidence in the prasugrel/VKA group was seen in the present study.

There were several differences in the results between AMI and non-AMI patients in the present study. In terms of mortality, the prasugrel/VKA group in non-AMI patients showed a significantly higher incidence of death compared with the clopidogrel/DOAC group. Although the mechanism of this finding is unclear, the combination of prasugrel and VKA should probably be avoided in non-AMI patients at low thrombotic risk. The incidence of myocardial infarction was decreased in the prasugrel/DOAC group in all patients and in AMI patients, but not in non-AMI patients. In AMI patients at high thrombotic risk, the forementioned strong and consistent antiplatelet inhibition with prasugrel may be useful to prevent recurrent myocardial infarction. Thus, low-dose prasugrel should probably be selected in AMI patients requiring triple therapy after PCI. A similar trend was also observed in the incidence of ischemic stroke, gastrointestinal bleeding, and blood transfusion between AMI and non-AMI patients.

In the present study, the frequency of patients receiving VKA as part of triple therapy was relatively high (44%). Previously, the ANAFIE registry, with more than 30,000 Japanese patients, reported that 28% of AF patients on anticoagulant therapy received VKA [31]. The present study included every patient on anticoagulant therapy. Therefore, some patients with specific conditions requiring VKA (e.g. mechanical valve, some types of thrombosis like left ventricular thrombosis, autoimmune disease, such as antiphospholipid syndrome) must have been included. This might be able to explain the high prevalence of VKA, although the database used in the present study did not have the information about actual indication for VKA or DOAC. Of note, VKA was consistently associated with a significantly higher rate of blood transfusion regardless of the type of P2Y12 inhibitor. Early switch from triple to dual

antithrombotic therapy, i.e. early cessation of aspirin, should be considered in patients requiring VKA.

## Limitations

There were several limitations in the present study. First, this retrospective study used the DPC administrative database, which did not have detailed clinical information including past medical history, laboratory data, and PCI strategies. Thus, residual confounding factors affecting clinical outcomes cannot be ruled out. Second, sample size calculation was not conducted for the primary outcome of in-hospital death because of the retrospective nature of this study. Third, clinical outcomes after discharge were not evaluated in this analysis. Fourth, there was no information detailing reasons for prescription of OAC (e.g. AF, deep vein thrombosis, or mechanical valve, etc.), which might influence the results in this study. Fifth, it is unclear whether all blood transfusions were needed due to bleeding events that occurred during the study period. Therefore, this outcome might not accurately represent bleeding events. Sixth, unstable angina was included in non-AMI patients in this study. Thus, some patients at high thrombotic/ischemic risk might be included in non-AMI patients. Seventh, the effect of another P2Y12 inhibitor, ticagrelor, could not be evaluated in this analysis, since ticagrelor is rarely used in Japan because of increased bleeding events reported in previous studies [32]. Finally, the database used in the present study did not include the time in therapeutic range in patients treated with VKA, which might be associated with clinical outcomes.

## Conclusions

Combination of low-dose prasugrel and DOAC was associated with lower incidence of MI, ischemic stroke, and blood transfusion compared with other regimens. Low-dose prasugrel may be feasible as part of triple therapy in patients requiring PCI and anticoagulation therapy.

## Acknowledgments

We thank Heidi N. Bonneau, RN, MS, CCA for her editorial review of the manuscript.

## Author Contributions

**Conceptualization:** Hideki Kitahara, Kazuya Tateishi, Yuki Shiko, Yusuke Inaba, Yoshio Kobayashi, Takahiro Inoue.

**Data curation:** Hideki Kitahara, Kazuya Tateishi, Yuki Shiko, Yusuke Inaba.

**Formal analysis:** Yuki Shiko, Yusuke Inaba.

**Supervision:** Yoshio Kobayashi, Takahiro Inoue.

**Validation:** Takahiro Inoue.

**Writing – original draft:** Hideki Kitahara.

**Writing – review & editing:** Kazuya Tateishi, Yuki Shiko, Yusuke Inaba, Yoshio Kobayashi, Takahiro Inoue.

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
