## [Decision Letter · Decision Letter 0]

17 Jun 2022

PONE-D-22-12588Comparison of short-term clinical outcomes between low-dose prasugrel and clopidogrel as part of triple antithrombotic therapy in patients requiring oral anticoagulant therapy and percutaneous coronary interventionPLOS ONE

Dear Dr. Kitahara,

Thank you for submitting your manuscript to PLOS ONE. After careful consideration, we feel that it has merit but does not fully meet PLOS ONE’s publication criteria as it currently stands. Therefore, we invite you to submit a revised version of the manuscript that addresses the points raised during the review process.

The reviewers have viewed your article in a mostly positive light, however still have some concerns. Please address these concerns in an individual fashion in your rebuttal letter. 

We look forward to receiving your revised manuscript.

Kind regards,

R. Jay Widmer

Academic Editor

PLOS ONE

Journal Requirements:

 [The funders had no role in study design, data collection and analysis, decision to publish, or preparation of the manuscript.]

Reviewers' comments:

Reviewer's Responses to Questions

**Comments to the Author**

1. Is the manuscript technically sound, and do the data support the conclusions?

Reviewer #1: Yes

Reviewer #2: Yes

2. Has the statistical analysis been performed appropriately and rigorously? 

Reviewer #1: Yes

Reviewer #2: I Don't Know

3. Have the authors made all data underlying the findings in their manuscript fully available?

Reviewer #1: Yes

Reviewer #2: Yes

4. Is the manuscript presented in an intelligible fashion and written in standard English?

Reviewer #1: Yes

Reviewer #2: Yes

5. Review Comments to the Author

Reviewer #1: In the ESC guideline, the combination of clopidgrel and DOAC as a part of triple antithrombotic therapy has been recommended to treat patients who need both OAC therapy and PCI procedure. However, in the guideline, the combination therapy of prasgrel and DOAC has not been indicated in such patients because of higher rates of adverse events including bleeding ones.

On the contrast, Japanese GL recommends 3.75mg/day doses of prasgrel in addition to ASA+DOAC to treat Japanese patients with same conditions. However, until now, there has been limited data to show the efficacies of the regimen.

The authors clearly showed safety and efficacy of combination of low-dose prasugrel and DOAC as a part of triple antithrombotic therapy, and the regimen was associated with lower incidence of events compared to combination of clopidgrel and DOAC.

The study included a large number of patients and the results are reasonable and promising and are of significance in the clinical practice.

The reviewer has some queries as shown below.

Specific comments

1. TTR was not evaluated in patients treated with VKA. This seems a limitation.

2. Please specify the dose of ‘standard-dose’ of prasugrel. But this may mean the dose for Western countries. Please add some explanation.

In addition, in Japan, much lower doses of prasugrel (2,5mg/day) may be used in patients with some specific conditions.

3. Did the authors include potassium-competitive acid blockers into PPI as the criteria? Please clarify.

PPI might be much more highly prescribed compared to Western countries. This may be one of the reason why such a low rate of GI bleeding was seen in the study. Please add some comments.

Reviewer #2: The study aims to look at low dose prasugrel as part of triple therapy.

1) What is the rational for the low dose prasugrel in Japan? You provide BMI but not weight. If this population weight less compared to western population, perhaps the low dose prasugrel is not low? Further, there is 5 mg available / used elsewhere

2) What is the reason for the high use of VKA? I understand you don’t have details, but this is quite high, as you mention in introduction that only certain patients such as mechanical valve require VKA

3) The study is likely not powered for primary outcome of death

4) What about outcome post discharge

5) The Figure 2 is hard to read due to poor resolution

6. PLOS authors have the option to publish the peer review history of their article (what does this mean?). If published, this will include your full peer review and any attached files.

Reviewer #1: **Yes: **Hideki ISHII

Reviewer #2: No

---

## [Author Response · Author response to Decision Letter 0]

7 Jul 2022

Reviewer #1: 

We would like to thank you for your careful review of our manuscript. We have revised our manuscript to incorporate your recommendations as fully as possible. Responses to the specific points are given below. 

Comment #1:

TTR was not evaluated in patients treated with VKA. This seems a limitation.

Authors’ Response to Comment #1:

Thank you very much for this important comment. In the revised manuscript, we have added the sentence “the database used in the present study did not include the time in therapeutic range in patients treated with VKA, which might be associated with clinical outcomes.” in the Limitation section (Page 14, Line 2-4).

Comment #2:

Please specify the dose of ‘standard-dose’ of prasugrel. But this may mean the dose for Western countries. Please add some explanation.

In addition, in Japan, much lower doses of prasugrel (2,5mg/day) may be used in patients with some specific conditions.

Authors’ Response to Comment #2: 

According to the reviewer’s suggestions, we have added the sentences explaining ‘standard-dose’ of prasugrel in Western countries, and ‘low-dose’ of prasugrel in Japan and some Asian countries in the Introduction section (Page 4, Line 12-13 and Line 17).

In addition, much lower dose of prasugrel (2.5 mg/day) was used in 0.1% of patients (4/2,815 patients) in this study. This result was described in the Results section (Page 7, Line 20-22).

Comment #3:

Did the authors include potassium-competitive acid blockers into PPI as the criteria? Please clarify.

PPI might be much more highly prescribed compared to Western countries. This may be one of the reason why such a low rate of GI bleeding was seen in the study. Please add some comments.

Authors’ Response to Comment #3: 

We appreciate this meaningful comment. In the present study, potassium-competitive acid blockers were included in PPI. The criteria were described in the Methods section in the revised manuscript (Page 6, Line 19).

In addition, the sentences regarding the low rate of GI bleeding were added in the Discussion section, as shown below.

“In terms of gastrointestinal bleeding, the incidence was quite low, and no significant difference was observed among the 4 subgroups. In this study population, PPI was administered in around 80-85% of patients. This high frequency of PPI administration might be associated with the reduced incidence of gastrointestinal bleeding.” (Page 11, Line 25 – Page 12, Line 4)

Thank you very much again for the time you have taken with our manuscript. We believe that incorporation of your suggestions significantly strengthened this manuscript.

Reviewer #2: 

We would like to thank you for your careful review of our manuscript. We have revised our manuscript to incorporate your recommendations as fully as possible. Responses to the specific points are given below. 

Comment #1:

What is the rational for the low dose prasugrel in Japan? You provide BMI but not weight. If this population weight less compared to western population, perhaps the low dose prasugrel is not low? Further, there is 5 mg available / used elsewhere

Authors’ Response to Comment #1: 

Thank you very much for the meaningful comments. Body weight in this population was approximately 63kg, which seems to be much lower than that in Western countries (Table 1).

When considering clinical introduction of prasugrel in Japan, special emphasis was placed on determining the dose to reduce the risk of bleeding events, because Japanese patients were considered to be older and have lower weight compared with Western patients. Dose-finding studies were carefully conducted, and randomized controlled trials confirmed the efficacy and safety of prasugrel 20/3.75 mg/day versus clopidogrel 300/75 mg/day in ACS patients as well as stable CAD patients undergoing PCI. These sentences were added in the Discussion section (Page 10, Line 18-23).

Comment #2:

What is the reason for the high use of VKA? I understand you don’t have details, but this is quite high, as you mention in introduction that only certain patients such as mechanical valve require VKA

Authors’ Response to Comment #2: 

We appreciate this important comment. In the present study, the frequency of patients receiving VKA as part of triple therapy was relatively high (44%). Previously, the ANAFIE registry, with more than 30,000 Japanese patients, reported that 28% of AF patients on anticoagulant therapy received VKA (Eur Heart J Qual Care Clin Outcomes. 2022;8:202-13). The present study included every patient on anticoagulant therapy. Therefore, some patients with specific conditions requiring VKA (e.g. mechanical valve, some types of thrombosis like left ventricular thrombosis, autoimmune disease, such as antiphospholipid syndrome) must have been included. This might be able to explain the high prevalence of VKA, although the database used in the present study did not have the information about actual indication for VKA or DOAC. These sentences were added in the Discussion section (Page 12, Line 24 – Page 13, Line 7).

Comment #3:

The study is likely not powered for primary outcome of death

Authors’ Response to Comment #3: 

We appreciate this important comment. In response, we have mentioned about this issue in the Limitation section: “sample size calculation was not conducted for the primary outcome of in-hospital death because of the retrospective nature of this study”. (Page 13, Line 16-18).

Comment #4:

What about outcome post discharge

Authors’ Response to Comment #4: 

Clinical outcomes post discharge were not evaluated in this analysis. Thus, we clearly mentioned about this issue in the Limitation section (Page 13, Line 18-19).

Comment #5:

The Figure 2 is hard to read due to poor resolution

Authors’ Response to Comment #5: 

Thank you for the comment. The resolution of Figure 2 was increased in the revised manuscript.

Thank you very much again for the time you have taken with our manuscript. We believe that incorporation of your suggestions strengthened this manuscript.

---

## [Decision Letter · Decision Letter 1]

14 Jul 2022

Comparison of short-term clinical outcomes between low-dose prasugrel and clopidogrel as part of triple antithrombotic therapy in patients requiring oral anticoagulant therapy and percutaneous coronary intervention

PONE-D-22-12588R1

Dear Dr. Kitahara,

We’re pleased to inform you that your manuscript has been judged scientifically suitable for publication and will be formally accepted for publication once it meets all outstanding technical requirements.

Kind regards,

R. Jay Widmer

Academic Editor

PLOS ONE

Additional Editor Comments (optional):

Reviewers' comments:

Reviewer's Responses to Questions

**Comments to the Author**

1. If the authors have adequately addressed your comments raised in a previous round of review and you feel that this manuscript is now acceptable for publication, you may indicate that here to bypass the “Comments to the Author” section, enter your conflict of interest statement in the “Confidential to Editor” section, and submit your "Accept" recommendation.

Reviewer #1: All comments have been addressed

Reviewer #2: (No Response)

2. Is the manuscript technically sound, and do the data support the conclusions?

Reviewer #1: Yes

Reviewer #2: Partly

3. Has the statistical analysis been performed appropriately and rigorously? 

Reviewer #1: Yes

Reviewer #2: I Don't Know

4. Have the authors made all data underlying the findings in their manuscript fully available?

Reviewer #1: Yes

Reviewer #2: No

5. Is the manuscript presented in an intelligible fashion and written in standard English?

Reviewer #1: Yes

Reviewer #2: No

6. Review Comments to the Author

Reviewer #1: Thank you for changings. The manuscript has been significantly improved. The reviewer has no criticism.

Reviewer #2: (No Response)

7. PLOS authors have the option to publish the peer review history of their article (what does this mean?). If published, this will include your full peer review and any attached files.

Reviewer #1: **Yes: **Hideki Ishii

Reviewer #2: No

---

## [Editor Report · Acceptance letter]

18 Jul 2022

PONE-D-22-12588R1 

Comparison of short-term clinical outcomes between low-dose prasugrel and clopidogrel as part of triple antithrombotic therapy in patients requiring oral anticoagulant therapy and percutaneous coronary intervention 

Dear Dr. Kitahara:

I'm pleased to inform you that your manuscript has been deemed suitable for publication in PLOS ONE. Congratulations! Your manuscript is now with our production department. 

Kind regards, 

on behalf of

Dr. R. Jay Widmer 

Academic Editor

PLOS ONE